# Empowering Efficient Spatio-Temporal Learning with a 3D CNN for Pose-Based Action Recognition

**DOI:** 10.3390/s24237682

**Published:** 2024-11-30

**Authors:** Ziliang Ren, Xiongjiang Xiao, Huabei Nie

**Affiliations:** 1School of Computer Science and Technology, Dongguan University of Technology, Dongguan 523820, China; renzl@dgut.edu.cn (Z.R.); xxiongjiang@outlook.com (X.X.); 2School of Artificial Intelligence, Dongguan City University, Dongguan 523419, China

**Keywords:** action recognition, 3D heatmap volumes, vision transformers (ViTs), pose modality, global cross learning

## Abstract

Action recognition based on 3D heatmap volumes has received increasing attention recently because it is suitable for application to 3D CNNs to improve the recognition performance of deep networks. However, it is difficult for models to capture global dependencies due to their restricted receptive field. To effectively capture long-range dependencies and balance computations, a novel model, PoseTransformer3D with Global Cross Blocks (GCBs), is proposed for pose-based action recognition. The proposed model extracts spatio-temporal features from processed 3D heatmap volumes. Moreover, we design a further recognition framework, RGB-PoseTransformer3D with Global Cross Complementary Blocks (GCCBs), for multimodality feature learning from both pose and RGB data. To verify the effectiveness of this model, we conducted extensive experiments on four popular video datasets, namely FineGYM, HMDB51, NTU RGB+D 60, and NTU RGB+D 120. Experimental results show that the proposed recognition framework always achieves state-of-the-art recognition performance, substantially improving multimodality learning through action recognition.

## 1. Introduction

Action recognition plays a crucial role in video understanding, and with the continuous development of sensor technology, previous research has investigated multiple modalities for feature representation, including RGB, skeleton, depth, and point-cloud modalities [1,2,3]. In particular, skeleton-based methods are concise and effective for action recognition due to their action-focusing nature and compactness. Recently, Graph Convolutional Networks (GCNs) have achieved remarkable success by capturing the interdependency among nodes through message passing within a graph structure [4,5]. However, it is difficult to reduce the impact of shifts in coordinate distributions of skeleton data in recognition tasks, and the inherent irregularity of skeletal graph structures impedes their direct amalgamation with modalities that are represented on uniform grids, such as RGB or depth modalities.

To overcome the above shortcomings, PoseConv3D [6] introduces a pose estimation method with 2D pose representations to facilitate multimodal feature fusion. A 2D pose is articulated through a series of heat maps, including joint and limb maps, as shown in Figure 1. Furthermore, PoseConv3D obtains 3D heatmap volumes by exporting data from joint or limb heat maps along the temporal axis. The pose sequence mainly includes joint coordinates and angular postures and is devoid of spatially inter-related positional data. Although PoseConv3D has demonstrated promising performance, it exhibits limitations in capturing global interdependencies. Vision Transformers (ViTs) with self-attention mechanisms achieve excellent performance in modeling inter-relations among input sequences but require a considerable parameter set and are prone to overfitting when confronted with intricate nuances.

More importantly, there is always a lack of spatio-temporal feature representation of video sequences in handling redundancy information of locally adjacent frames or capturing complex dependencies between distant frames. To address this issue, some frameworks have been designed to improve the feature-learning ability and enhance video classification performance, such as 3D CNNs [7,8,9] and spatio-temporal transformers [10]. Unfortunately, each of these frameworks focuses on a specific challenge. By utilizing context from a small 3D neighborhood (e.g., 3×3×3), a 3D CNN effectively captures intricate and localized spatio-temporal features, thereby mitigating redundancy among neighboring frames. Nevertheless, the effectiveness of this method in modeling long-range dependencies is hindered by a constrained receptive field  [11,12]. On the other hand, spatio-temporal transformers [13] excel in capturing global dependencies. However, using UniFormer [14], it was found that in early network layers, the spatial attention of TimeSformer [13] is mainly focused on neighbor tokens, and its temporal attention tends to only focus on tokens in adjacent frames. To tackle these difficulties, UniFormer proposes a relation aggregator to learn local relations through a small convolutional network in the shallow layer. Similarly, the aggregator can learn global relations through self-attention mechanisms in deep layers, flexibly constructing long-range token dependencies. However, the model requires tedious and complex image pre-training before fine tuning on videos. In addition, when combined with a self-attention mechanism, this model must utilize a non-hierarchical convolutional network.

Thus, inspired by the 3D heatmap volumes of base representations and self-attention mechanisms, to better represent spatio-temporal features and fully utilize multimodal information, we propose a novel pose-based action recognition model that seamlessly combines the advantages of 3D convolution and spatio-temporal self-attention in a concise format, thereby achieving a desirable balance between computation and accuracy. The proposed recognition framework consists of three stages and Global Cross Blocks (GCBs), which are designed to build long-range token dependencies from intermediate feature sequences of convolutional networks combined with class embeddings. The main contributions of this paper are summarized as follows:We propose an effective ConvNet with global cross blocks to learn local relations and build long-range token dependencies from intermediate features using shallow layers. The designed GCBs consist of four core modules: Dynamic Position Embedding (DPE), a Multi-Head Relational Aggregator (MHRA), a Feedforward Network (FFN), and Average Class Prediction (ACP), which work collaboratively to enhance performance in action recognition.For the proposed recognition framework, we design an improved Global Cross Complementary Block (GCCB) to promote early-stage feature fusion between RGB and asymmetric pose pathways, which can be easily integrated in multi-stream networks to learn compensation features from different modalities.To evaluate the effectiveness of the proposed recognition framework, we conducted a series of comprehensive experiments on four widely used video benchmarks, including FineGYM [15], HMDB51 [16], and NTU RGB+D 60&120 [17,18]. Our results demonstrate that the proposed models achieved state-of-the-art recognition performance for all benchmarks.

The rest of this paper is organized as follows. Section 2 discusses related work on RGB- and skeleton-based action recognition methods. In Section 3, we describe the details of the proposed PoseTransformer3D recognition framework. Section 4 presents the implementation details and a comparison with current state-of-the-art methods, and Section 5 concludes the paper.

## 2. Related Works

In this section, we briefly review recent works related to RGB- and skeleton-based human action recognition, including Convolutional Neural Networks (CNNs), Recurrent Neural Networks (RNNs), Graph Convolutional Networks (GCNs), Graph Neural Networks (GNNs), and transformers.

RGB-based action recognition. As the most easily obtainable common data modality, RGB data encompass abundant appearance and scene information. Therefore, RGB-based models of action recognition have been rapidly developed and well applied and can be broadly classified into three categories: two-stream ConvNets, 3D CNNs, and transformer-based methods.

Simonyan et al. [19] first introduced a classic two-stream ConvNet to extract spatial and temporal features from RGB and optical flow modalities. Their model has been extended by several researchers. Based on the observation of the existing two-stream CNN’s relatively shallow layers, Wang et al. [20,21] proposed a segmented sampling mechanism to extract long-term video-level information for action recognition and designed a deeper version to achieve improved recognition results. Meanwhile, Zhang et al. [22] proposed a teacher–student framework to tackle the computational complexity associated with accurate optical flow computation. Moreover, there have been other variations of the two-stream CNN architecture. To enhance the feature representation of irrelevant frames, Adascan et al. [23] recursively predicted the discriminative importance of each frame to guide feature pooling for action recognition.

Compared with traditional 2D CNNs, 3D CNNs have the advantage of capturing temporal dynamics in video sequences, thereby achieving more accurate action classification. To optimize computational efficiency and reduce the number of parameters, researchers have proposed improved and variant 3D CNN models [11,24,25,26]. I3D [24] was designed with two-stream 3D convolutional networks and uses 2D convolution to process the temporal dimension to reduce the parameters of the 3D CNNs, achieving remarkable results in terms of both accuracy and efficiency. In addition, several advanced deep architectures have been proposed to further enhance performance, among which the typical network, Slowfast [8], introduces a two-stream 3D convolutional framework that separately processes low-frequency and high-frequency RGB video frames to capture semantic and motion information.

Furthermore, transformer models with self-attention mechanisms exhibit exceptional competence in long-term dependencies and capturing spatio-temporal relationships within video frames [10,13,14,27,28,29]. The authors of [10] decomposed an input video along spatial and temporal dimensions to extract spatio-temporal tokens, which were then encoded by a series of transformer layers to handle long token sequences encountered in the video. TimeSformer [13] was devised with a self-attention mechanism that applies the standard transformer architecture to focus on learning spatio-temporal features in a sequence of frame-level patches. These studies incurred considerable computational costs and lack a hierarchical structure, which may severely limit the action recognition performance when handling voluminous 3D video data. To respond to these issues, Uniformer [14] uses a 3D convolutional network with a spatio-temporal self-attention mechanism to establish long-range token dependencies across frames in a video and reduce the computational burden. Fan et al. [27] proposed a hierarchical transformer network (MViT) by combining a multiscale pyramid of features with the transformer model, while VideoSwin [28] comprises a hierarchical architecture to confine computations by using a 3D-shifted window and a multi-head self-attention module. Incorporating local window attention to encode detailed spatio-temporal information and global attention modules, Weng et al. [29] designed a hierarchical spatio-temporal pyramid transformer model to capture long-term spatio-temporal dependencies.

In contrast to the aforementioned approaches, our model integrates the merits of 3D convolution and spatio-temporal self-attention within a compact ResNet framework, which allows us to strike a desirable equilibrium between computational efficiency and accuracy.

Skeleton-based action recognition. A skeleton sequence exhibits scale invariance and robustness to texture and background variations. Many related deep convolutional network models have been designed for action recognition.

A skeleton sequence is usually divided into five body parts, then input into the designed networks to implement classification tasks. Du et al. [30] proposed a classic hierarchical RNN for analyzing human skeleton data. Their model can effectively extract spatio-temporal features in both directions. Zhu et al. [31] designed a P-LSTM network to model the relationships between different body parts that can better understand the interdependencies between different body parts than previous models. Furthermore, Song et al. [32] presented a deep LSTM architecture with spatio-temporal attention that incorporates both spatial and temporal attention subnetworks into the main LSTM network to capture complex spatio-temporal features.

Due to the advantages of skeleton sequences and the high-performance characteristics of graph convolutional neural networks, GCNs have been developed rapidly for action recognition tasks [33,34]. GCN-based methods can effectively handle human skeleton sequences with complex relational structures of nodes and edges. Yan et al. [33] constructed a spatio-temporal graph network, ST-GCN, which employs a two-stream network to model spatio-temporal relationships for skeleton-based action recognition. Chen et al. [34] proposed Channel-wise Topology Refinement Graph Convolution (CTR-GC) for skeleton-based action recognition. Their model can effectively capture the intrinsic correlation between joints.

Although GCN-based methods have achieved great success in skeleton-based action recognition, they still have some limitations in recognizing fused skeletons and other modality features [6]. To fully extract the features of a skeleton sequence, an effective method is to design a transformation module to represent it as a pseudo image; some researchers have suggested directly converting the coordinate information into pseudo-images using transformation modules [35,36,37]. However, compared to GCN-based methods, these methods do not fully utilize the advantages of convolutional networks and, therefore, do not prove particularly advantages on commonly used benchmarks [35]. Inspired by 3D heatmap volume representation and global attention mechanisms, in this paper, we design a 3D deep neural networkframework to learn local and global features for pose-based action recognition.

## 3. PoseTransformer3D Framework

In this paper, we design a novel model, namely, PoseTransformer3D, to improve the spatio-temporal modeling of skeleton sequences and provide RGB and skeleton modality features within a multi-level fusion framework, RGB-PoseTransformer3D, to improve recognition performance.

### 3.1. PoseTransformer3D

To capture local features and long-term global spatio-temporal dependencies from a skeleton sequence a novel 3D convolutional structure with a global cross block based on PoseConv3D [6] is designed as the backbone. Due to the resolution of 3D heatmap volumes being reduced by four times relative to that of RGB images, early downsampling operations can be removed from th e3D CNN to meet the input requirements of deep network models. Therefore, we first use RGB and skeleton sequences to generate 3D heatmap volumes and input them into the network model for feature extraction. Then, to address the problem of integrating a 3D CNN, our model proposes a hierarchical transformer with a global attention mechanism that progressively shrinks the spatio-temporal resolution of feature maps while expanding channels as the network gets deeper. In addition, we pre-train the model using a 3D CNN, then fine tune it, resulting in better performance compared to training the model directly. The details of the proposed PoseTransformer3D framework are shown in Figure 2.

For RGB videos and skeleton sequences, we first extract the 2D human pose and apply pre-processing techniques to obtain 3D heatmap volumes, then utilize designed the PoseTransformer3D model to achieve action classification. In the process of feature learning, the 3D CNN is used to eliminate the early downsampling operation and capture the spatio-temporal dynamics of 3D heatmap volumes. The designed GCB consists of three essential components: Dilated positional encoding (DPE), a multi-head relation aggregator (MHRA), a and feed forward network (FFN).

Our recognition framework integrates a global cross-attention mechanism into the 3D CNN and represents the multi-scale spatio-temporal features in token form to enhance discriminative representations. In addition, we add positional embedding and learnable queries (*q*) to capture the relative positional relationships between different elements in sequence data. The distinct design of our PoseTransformer3D model is summarized in Table 1.

### 3.2. Global Cross Block

Vision transformers can effectively capture long-range dependencies through self-attention mechanisms. However, they may suffer from computational redundancy due to the blind similarity comparisons across all tokens in each layer. As a result, the model may encounter a substantial computational load when encoding local video representations in the shallow layers, resulting in unsatisfactory accuracy and efficiency in spatio-temporal learning. To address this issue, our PoseTransformer3D model learns local relations through a convolutional network in the shallow layer. In the deep layer, we design and integrate a GCB with a hierarchical structure into the convolutional network, which consists of an L-layer global transformer decoder, a fully connected layer, ACP and classification loss, as shown in Figure 2. The global transformer decoder consists of three essential parts—DPE, MHRA, and FFN—which are defined as follows:(1)$$X^{L}=DPE\left(X_{in}\right)+X_{in},$$
(2)$$X^{ST}=C_{MHRA}\left(Norm\left(q\right),Norm\left(X^{L}\right)\right),$$
(3)$$X_{out}=FFN\left(Norm\left(X^{ST}\right)\right)+X^{ST},$$
where $X_{in}$ is the input of the GCB; $X^{L}$ and $X^{ST}$ represent intermediate features learned through DPE and MHRA; and Norm(·) and $C_{MHRA}\left(\cdot \right)$ indicate the regularization and concatenation operations, respectively. DPE [14] is an expanded positional encoding method used to capture long-distance dependencies. MHRA(·) is a attention mechanism used to construct representations between different positions. The MHRA can be formulated as follows:(4)$$A_{n}^{C}\left(q,X_{j}^{L}\right)=\frac{e^{\left(Q_{n}{\left(q\right)}^{T}K_{n}\left(X_{j}^{L}\right)\right)}}{\sum_{j^{\prime }\in \Omega_{T\times H\times W}}e^{\left(Q_{n}{\left(q\right)}^{T}K_{n}\left(X_{j}^{L}\right)\right)}},$$
where the cross affinity matrix ($A_{n}^{C}\left(\cdot \right)$) is computed to capture the relationship between queries and $X^{L}$.
(5)$$R_{n}^{C}\left(q,X^{L}\right)=A_{n}^{C}\left(q,X^{L}\right)V_{n}\left(X^{L}\right),$$
(6)$$C_{MHRA}=Concat\left(R_{1}^{C},R_{2}^{C},\cdots ,R_{N}^{C}\right)U,$$
where $R_{n}^{C}\left(\cdot \right)$ can transform a learnable queries ($q\in R^{N\times C}$) into a video representation by modeling the dependencies between queries and all spatio-temporal tokens ($X^{L}$). Note that query tokens are initialized to zero to ensure stable training, and *C* is the number of channels of spatio-temporal tokens ($X^{L}$). Then, a linear projection is applied to transform *X* into a spatio-temporal context ($V_{n}$). Subsequently, $V_{n}$ is aggregated into a learnable query, guided by the affinity ($A_{n}^{C}\left(q,X^{L}\right)$). Finally, query tokens enhanced by all the heads are fused together to form the final video representation using a linear projection ($U\in R^{C\times C}$). FFN contains multiple hidden layers and activation functions that can capture more complex nonlinear relationships in the input sequence.

Relative positional embedding is a technique for capturing the relative positional relationships between different elements in sequence data, which can help models better understand the context and semantic information of sequence data. As demonstrated in [38], we encode the relative position between two input elements (p(i) and p(j)) as a positional embedding ($R_{p(i),p(j)}\in R^{d}$, where p(i) and p(j) denote the spatial (or spatio-temporal) position of elements *i* and *j*, respectively). Equation (4) is reformulated as
(7)$$A_{n}^{C}\left(q,X\right)=Softmax\left(Q_{n}{\left(q\right)}^{T}{\;K}_{n}\left(X\right)+E^{\left(rel\right)}\right),$$
where $E_{ij}^{\left(rel\right)}=Q_{i}\cdot R_{p(i),p(j)}$. To reduce complexity, we decompose the distance calculation between element *i* and element *j* along the spatio-temporal axis as follows:(8)$$R_{p(i),p(j)}=R_{h(i),h(j)}^{h}+R_{w(i),w(j)}^{w}+R_{t(i),t(j)}^{t}.$$

To generate a more discriminative video representation, we adaptively integrate spatio-temporal features from $X_{in}$, and $X_{out}$ with *N* queries can be obtained. Then, linear projection is used to transform $q_{i}$ into the predicted value of action recognition ($p(c|q_{i})$):(9)$$p\left(c|q_{i}\right)=\log\frac{e^{\left(W_{c}q_{i}+b_{c}\right)}}{\sum_{c_{i}=1}^{C}e^{\left(w_{c_{i}}q_{i}+b_{c_{i}}\right)}},$$
where *C* represents the number of action categories and $W_{c}$ and $b_{c}$ correspond to the weight and bias of the softmax layer, respectively. Finally, ACP is used to average multiple prediction results to obtain the final prediction result, which can be formulated as follows:(10)$$ACP=\frac{1}{N}\sum_{i=1}^{N}p\left(c|q_{i}\right),$$
where p(·) represents each element in the prediction result and *N* denotes the total number of elements in the tensor. The loss functions for different GCB are optimized as follows:(11)$$L_{Global}\left(y,C\right)=-\sum_{c=1}^{C}y_{c}\left(ACP(q)\right),$$
where $y_{c}$ is the ground-truth label for action *c*. In the training process, we mainly consider the loss function from both the 3D CNN and GCB, which can be formulated as follows:(12)$$L=\lambda_{3D-CNN}L_{3D-CNN}+\lambda_{GCB}\sum_{m=1}^{M}L_{Global}{\left(y,C\right)}_{m},$$
where $L_{CNN}$ and $L_{Global}\left(y,C\right)$ corresponding to the 3D CNN and GCB, respectively; *M* is the number of GCBs; and $\lambda_{CNN}$ and $\lambda_{GCB}$ are hyperparameters to balance the contribution of each task. Furthermore, to complement the advantages of different modules, we integrate results as follows:(13)$$F_{score}=Fusion\left(F_{GCB},F_{CNN}\right),$$
where Fusion(·) is an average ensemble operation and $F_{GCB}$ and $F_{CNN}$ represent the scores of GCB processing and the convolutional network, respectively.

### 3.3. RGB-PoseTransformer3D

To explore the performance of PoseTransformer3D, we designed an improved variant, RGB-PoseTransformer3D, for multimodality fusion action recognition. The improved RGB-PoseTransformer3D can simultaneously learn features from different modalities and capture global dependencies using an added self-attention mechanism.

As shown in Figure 3, we designed two asymmetric pathways to handle the distinct characteristics. The RGB pathway possesses a larger channel width, greater depth, and higher-spatial-resolution input. Furthermore, a global cross complementary block (GCCB) is designed to promote early-stage feature fusion between the two modalities in GCB networks, as shown in Figure 4. $P_{token}$ and $RGB_{token}$, with different channel numbers from the output of the first GCB, are connected by a residual pooling connection. The output of the residual pooling connection is linearly projected to match the dimensions of the model and combine the pose and RGB feature sequences, forming the input tokens for the second GCB. $P_{embeddings}$ and $RGB_{embeddings}$ indicate class embeddings (queries (*q*)) with relative positional embedding. ⊕ represents the residual pooling connection.

## 4. Experimental Results and Analysis

### 4.1. Implementation Details

3D CNN Backbone. In experiments, we chose SlowOnly [8] as the default backbone due to its simplicity (inflated directly from ResNet) and impressive recognition performance. As shown in Table 2, the pose pathway has a higher frame rate and a smaller channel width, while the RGB pathway has a lower frame rate and a larger channel width.

Global transformer decoder. Based on image features *F* and *N*, a global transformer decoder is utilized to compute position embeddings (queries (*q*)) to encode the *F* predicted by the 3D CNN. We assign queries a corresponding relative positional embedding and utilize one global transformer decoder layer with one query.

Training settings. The pre-training technique used in MMaction2 [39] was applied in our experiments, and the stochastic gradient descent (SGD) and CosineAnnealingLR mechanisms were used to optimize the proposed networks with an initial learning rate of 0.005 and a weight decay of 0.0001. Unless otherwise specified, we utilize a crop size of 56 × 56 for the pose sequence and a crop size of 224 × 224 for the RGB sequence. In addition, the data augmentation techniques used in the experiments are consistent with those employed in [6]. We maintain a batch size of 16 and train all models for 60 epochs. In both late and early fusion + late fusion approaches, we combine predictive values from the two modalities in a 1:1 ratio to obtain the final prediction.

Network inputs. Our model utilizes RGB and pose data from videos to implement an action recognition classification. Initially, we employ a segment and uniform sampling strategy to choose a sequence of frames within the video. To extract the 2D human pose from each frame, we employ a two-stage pose estimator consisting of both detection and pose estimation, as proposed in [6]. Subsequently, the heat maps of joints or limbs are stacked along the temporal dimension and preprocessed to generate 3D heatmap volumes. Ultimately, we employ PoseTransformer3D to classify these volumes. More information about the algorithm can be found in Algorithm 1.
**Algorithm 1** Details of pose-based action recognition framework with PoseTransformer3D**Input:** VRGB, VSkeleton**Output:** Action recognition classification.
  1:Stage1: Generated 3D heatmap volumes  2:**for** each RGBandSkeleton **do**  3:   **for** each SegmentS **do**  4:     **for** each SamplingT **do**  5:        V2DPose(T)=fDE(VRGB(T),VSkeleton(T))← Detection + Pose estimation  6:     **end for**  7:     V3Dheatmapvolumes=fSP(V2DPose)← Stack + Preprocessing  8:   **end for**  9:**end for**10:Stage2: Pre-training a 3D-CNN for classifying 3D heatmap volumes11:**for** each SamplingS **do**12:   p(c∣V)=logeWcV+bc∑ci=1CewciV+bci← Features extraction with 3D-CNN13:**end for**14:Stage3: PoseTransformer3D classifies 3D heatmap volumes based on 3D-CNN pre-training15:**for** each SamplingS **do**16:   p(c∣V)3D−CNN=logeWcV+bc∑ci=1CewciV+bci← Features extraction with 3D-CNN Backbone17:   p(c∣q)GCB=logeWcq+bc∑ci=1Cewciq+bci← Features extraction with GCB18:   p(c∣(V,q))=λ3D−CNNp(c∣V)3D−CNN+λGCB∑m=1Mp(c∣q)GCB← Features extraction with PoseTransformer3D19:**end for**20:**return** action classification


### 4.2. Datasets

HMDB51 [16] is a database of 7000 video clips covering 51 common human actions. It is a widely used dataset in the field of action recognition research, consisting of real videos uploaded by users that are close to real-life scenarios. It contains 101 action categories, and each category contains approximately 100 clips, covering a total of 13,320 video clips.

NTU RGB+D [17,18] is a multimodal dataset for action recognition and analysis, including NTU RGB+D 60 and NTU RGB+D 120. NTU RGB+D 60 captures human behaviors across 60 action categories and employs two evaluation protocols, namely cross-subject (C-Sub) and cross-view (C-View) protocols. NTU RGB+D 120 provides more than 114,000 video samples and 8 million frames jointly captured by three cameras in different orientations under 32 settings, covering 106 different subjects and 120 action categories, and contains two access protocols: cross-subject (X-Sub) and cross-set (X-Set) protocols.

FineGYM [15] is a fine-grained action recognition dataset that is a challenging benchmark with high-quality annotations. It comprises 29K videos of 99 fine-grained gymnastic action classes. These video clips are divided into a three-level hierarchy (namely events, sets, and elements) and two temporal levels (namely action and sub-action). Regarding pose extraction, we utilize human poses extracted with ground-truth bounding boxes for the athlete in all frames.

### 4.3. Properties of PoseTransformer3D

We compared PoseTransformer3D with MS-G3D [40] and Pose-SlowOnly [6]. To ensure a fair performance comparison, we adopted an input shape of 48×56×56 for PoseTransformer3D. After the res3 and res4 stages, we employed a GCB consisting of 1 global transformer decoder layer with a one query, the results of which are shown in Table 3. The results show that the proposed PoseTransformer3D achieves competitive performance across various datasets. Moreover, PoseTransformer3D is the same as PoseConv3D, using the same parameters and FLOPs.

To test the generalization of PoseTransformer3D, we conducted extensive experiments on four datasets. PoseTransformer3D adopts the 3D heatmap volumes introduced in PoseConv3D, which consist of joint-based heat maps and limb-based heat maps. Here, 3D heatmap volumes with dimensions of 48×56×56 are utilized as input; then, the accuracy obtained through 1-clip and 10-clip testing are reported. Table 4 presents a comparison of PoseTransformer3D with PoseConv3D for skeleton-based action recognition. The results indicate that PoseTransformer3D achieves comparable recognition performance to that of PoseConv3D in 1-clip testing. Moreover, PoseTransformer3D consistently outperforms the current state-of-the-art PoseConv3D in 10-clip testing, showing its superiority in terms of recognition. Despite PoseTransformer3D’s strengths, the computational complexity of the model is significant, with increased parameters and FLOPs that may limit real-world deployment, particularly in resource-constrained environments. To effectively address this challenge, we can adopt a model-pruning strategy to reduce the number of parameters and computation. Specifically, we optimized the number of convolutional network layers from four to three. At the same time, the GCB network was also adjusted to adopt a single-layer self-attention mechanism. These improvements significantly reduced the complexity of the model and improved its feasibility in practical applications.

### 4.4. RGB-PoseTransformer3D

The proposed RGB-PoseTransformer3D can learn complementary features from different modalities. We conducted a comparison between two single-modality and multimodal late-stage fusion methods, the results of which are shown in Table 5. The results demonstrate that the multimodal late fusion is greatly improved after being combined with RGB and pose features, which indicates that improved performance can be attained during the testing phase by fusing features from different modalities.

RGB-PoseTransformer3D enhances flexibility in fusing pose with RGB features using early fusion strategies. Specifically, we utilize lateral connections between the RGB pathway and the Pose pathway to fuse features across the two modalities at an early stage. By incorporating both early and late fusion, RGB-PoseTransformer3D achieves superior performance. The results presented in Table 6 show the performance improvement achieved by early + late fusion of both pose and RGB volumes. The results achieved by bi-directional feature fusion in the early stage using RGB and pose features with 1-clip/10-clip testing are close to those achieved using early + late fusion with 1-clip/10-clip testing. Furthermore, compared with single-modality RGB and pose methods, RGB-PoseTransformer3D can achieve better fusion results in early + late fusion.

### 4.5. Comparisons with the State of the Art

Skeleton-based Action recognition. Table 7 presents a comparison between PoseTransformer3D and previous studies on skeleton-based action recognition. Compared to a 3D skeleton, PoseTransformer3D obtained the current state-of-the-art results by utilizing improved 3D heatmap volumes. For joint = (J) and limb (L)-based heatmap fusion strategies, PoseTransformer3D achieves the best performance, which can be attributed to the excellent performance of PoseTransformer3D on a single modality. Furthermore, PoseTransformer3D utilizes input that consists of 3D heatmap volumes with dimensions of 48×56×56, and the reported accuracy is based on a 10-clip testing approach.

Multimodality fusion. By utilizing multimodality fusion, RGB-PoseTransformer3D achieves outstanding results across four different benchmark datasets. Table 8 demonstrates that our early + late fusion consistently delivers exceptional performance. Furthermore, compared to PoseConv3D [6], the proposed RGB-PoseTransformer3D with a self-attention mechanism achieves even better results.

### 4.6. Visualization

To visualize the PoseTransformer3D model, we utilized Grad-CAM [52] to compute the learned attention maps. Figure 5 shows a test attention image of the convolutional component (res4) and DPE. It can be observed that the convolutional component focuses on local regions, particularly regions with motion, while the DPE pays more attention to global regions. Furthermore, we find that there is a certain complementarity between the two attention maps.

### 4.7. Ablation Study

Generic pre-training paradigm. Transformer models usually require a large amount of training data to achieve good performance, while 3D CNNs always utilize weight sharing and pooling to extract local features. To improve recognition performance, a generic pre-training paradigm is used to combine the ability of transformers with the proposed model. Pose-Transformer requires a tedious and complex image pre-training phase before it can be fine-tuned to adapt to video processing tasks. This blocks its widespread usage in practice. In contrast, open-sourced CNNs are readily available and well pre-trained with rich image supervision. Based on these observations, we propose a generic paradigm to build our networks by combining a pre-trained CNN with an efficient GCB. Pose-Transformer can achieve faster convergence while requiring fewer computing resources.As shown in Table 9, the pre-training mechanism achieves a noteworthy enhancement in recognition effectiveness, which may be due to the fact that pre-training introduces more prior knowledge or heuristic information during the training process.

Number of global transformer decoder layers. We specifically isolate the component of global transformer decoder layers while disregarding other influences from the experimental environment. As shown in Table 10, a GCB with only one transformer decoder layer already excels in action recognition, but increasing the number of decoder layers becomes more crucial when it comes to differentiating highly similar actions. The reason is that stacking additional decoder layer aids in eliminating redundant predictions, which is essential for accurately distinguishing between highly similar actions. Figure 6 shows the improvement of PoseTransformer3D with one transformer decoder layer over no self-attention method, indicating that the former can learn global features from the GCB.

Number of queries. Parameter queries (*q*) govern the weights of the relative positional features between different elements in sequence data. Intuitively, the value of *q* affects the contribution of modalities by adding a relevant mechanism. We attempted to vary *q* from 0 to 100 to verify the best performance of PoseTransformer3D; the comparison results are shown in Table 11. Note that q=0 indicates that no self-attention or GCB is adopted. A larger *q* value only slightly improves the outcome; thus, from a comprehensive perspective, there is no need to increase the number of queries.

To further investigate the performance of our proposed model, the experimental results are comprehensively analyzed, and the confusion matrix of our model with a GCB (q=1 query) on the FineGYM dataset are shown in Figure 7. Obviously, our PoseTransformer3D achieves both more accurate predictions and fewer misclassifications, which is attributed to the fact that the GCB learns more global information from the 3D heatmap volumes. In Figure 7, we provide a visualization of the prediction accuracy for each category. We qualitatively analyzed the error categories and found that our model is limited in its ability to distinguish between highly similar actions.

Global cross blocks and multimodality fusion. In Table 12, we find that self-attention is better than convolution for deep layers and that our Global MHRA is more powerful than both of them on the FineGYM dataset. This results reveals that the features in the deep layers are critical for capturing long-term dependency, while the DPE and the middle information are necessary for identifying the motion difference. For the fusion strategy, the GCCB is designed to facilitate early-stage feature fusion. Table 13 demonstrates that simple late fusion is sufficient for the integration of multi-stage features. We find that late + GCCB fusion can significantly improve recognition accuracy.

## 5. Conclusions

In this paper, we propose a novel PoseTransformer3D method and extend its implementation to multimodal action recognition. The designed model utilizes a heatmap volume mechanism to represent the spatio-temporal information from RGB and skeleton sequences. A 3D convolution with a GCB and a self-attention module is also developed to extract discriminative features from 3D heatmap volumes. The GCB is used not only to learn local relations but also to build long-range token dependencies from intermediate features. Experimental results verify the effectiveness of the proposed PoseTransformer3D and RGB-PoseTransformer3D. Compared with existing pose-based methods, the proposed framework effectively leverages pose information and strikes a desirable balance between accuracy and efficiency. However, in this paper, we only conducted testing on public datasets. In the future, the design of lightweight models and testing in real-world scenarios can help achieve more efficient and practical value.

## Figures and Tables

**Figure 1 sensors-24-07682-f001:**
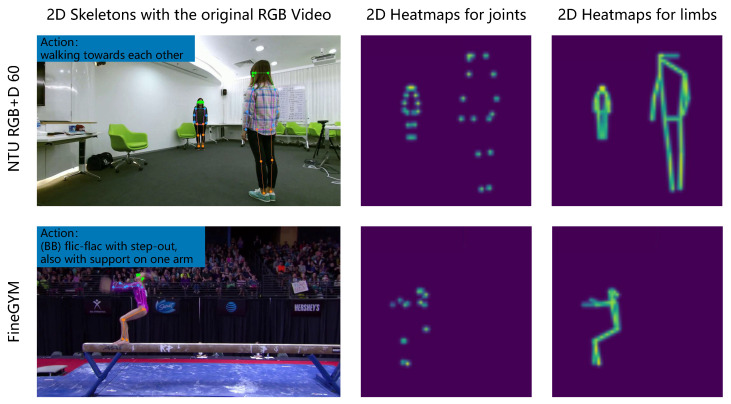
Visualized content from the NTU RGB+D 60 and FineGYM datasets, which include “2D Skeletons with the original RGB Video”, “2D Heatmaps for joints”, and “2D Heatmaps for limbs”.

**Figure 2 sensors-24-07682-f002:**
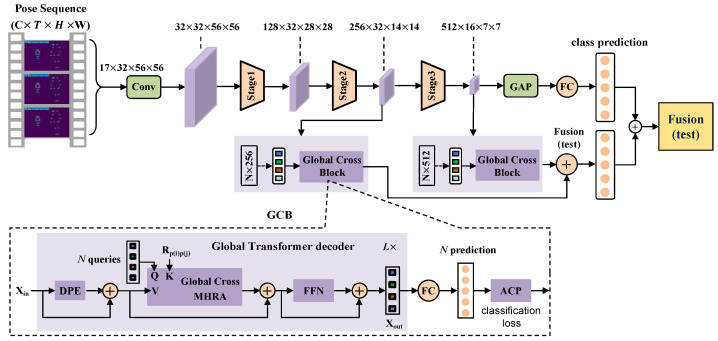
Overview of the proposed PoseTransformer3D model. $R_{p(i),p(j)}$ denotes the relative positional embedding, ⊕ represents residual pooling connection, L× indicates the length of the decoder, and $X_{in}$ represents the intermediate features of the 3D CNN network.

**Figure 3 sensors-24-07682-f003:**
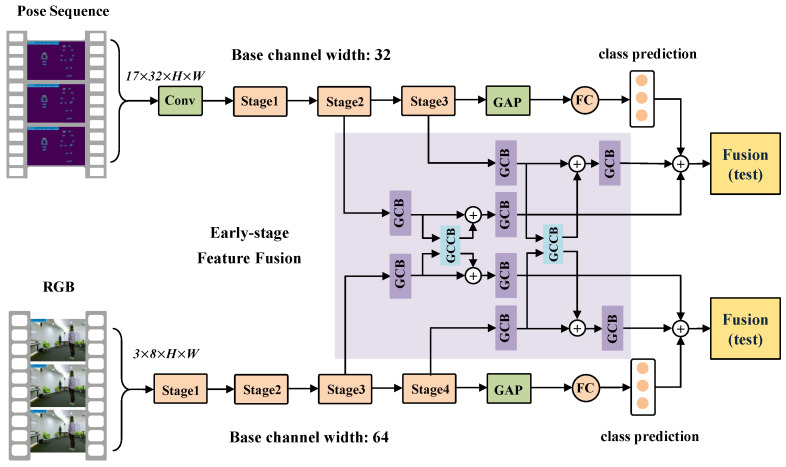
Overview of the proposed RGB-PoseTransformer3D model. The input sequence consists of RGB and pose modalities, which are then projected into an asymmetric two-stage 3D CNN pathway and subjected to early-stage feature fusion. We use self-attention to model intermediate features for asymmetric two-stage 3D CNN pathways, with cross-modal information flow implemented via the global cross complementary block (GCCB) in the network.

**Figure 4 sensors-24-07682-f004:**
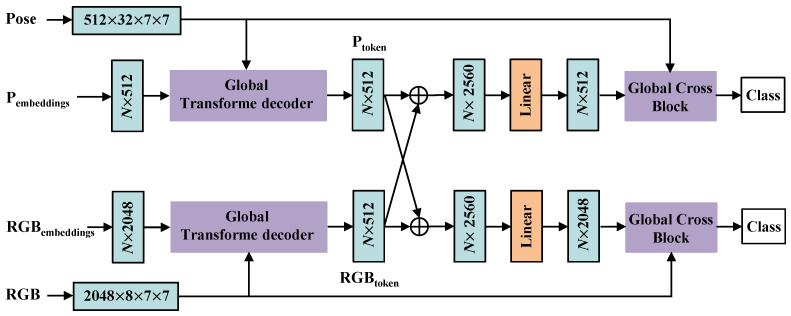
Overview of the global cross complementary block (GCCB). The input tokens for the first GCB are composed of pose and RGB features with class embeddings.

**Figure 5 sensors-24-07682-f005:**
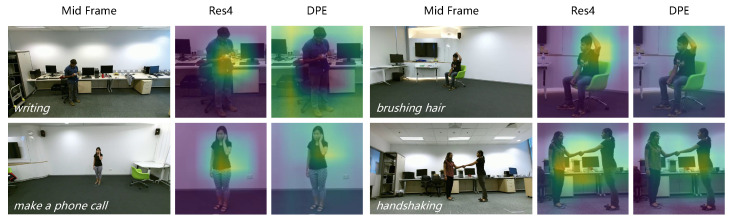
Attention maps of several action samples from the NTU RGB+D 60 database. The original middle frame of each video clip is displayed on the left side. DPE illustrates the attention maps computed by a component in the second global cross blocks (GCB), while Res4 represents the attention maps generated by the convolutional component (ResNet50 res4).

**Figure 6 sensors-24-07682-f006:**
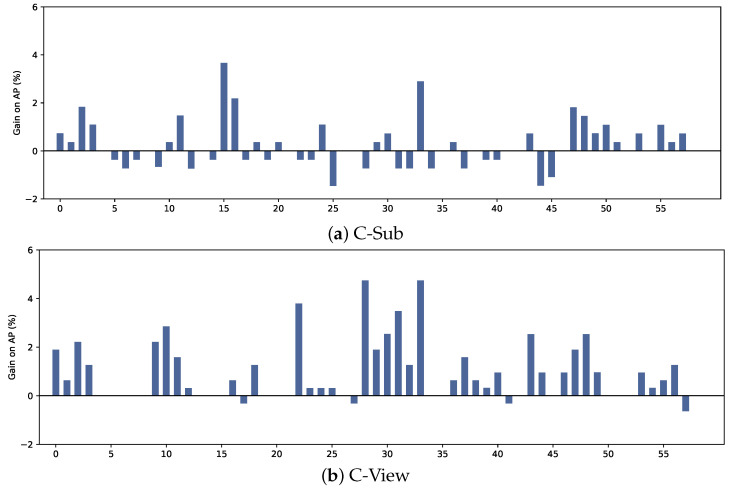
The average precision gain of Pose-Transformer3D with one Transformer decoder layer compared to no self-attention method on the NTU RGB+D 60 dataset. The vertical axis represents the percentage of average precision gain, and the horizontal axis represents 60 action IDs.

**Figure 7 sensors-24-07682-f007:**
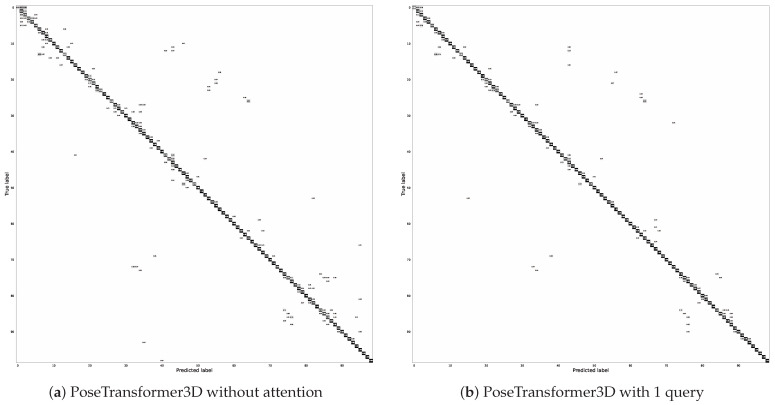
Confusion matrices of PoseTransformer3D on the FineGYM dataset.

**Table 1 sensors-24-07682-t001:** Differences between PoseConv3D and PoseTransformer3D.

Method	PoseConv3D	PoseTransformer3D
Input	2D Skeleton	2D Skeleton
Format	3D heatmap volumes	3D heatmap volumes
Architecture	3D CNN	3D CNN + GCB
Queries	N.A.	Learnable queries

**Table 2 sensors-24-07682-t002:** The kernel dimensions are represented as $T\times S^{2}\times C$, where *T* represents the temporal size, $S^{2}$ represents the spatial size, and *C* represents the channel size. Temporal and spatial strides are denoted as *T* and $S^{2}$, respectively. We employ ResNet50 as our backbone.

Stage	RGB Pathway	Pose Pathway	Output Sizes ($T\times S^{2}$)
data layer	uniform 8, $1^{2}$	uniform 32, $4^{2}$	RGB: $8\times 224^{2}$ Pose: $32\times 56^{2}$
stem layer	conv $1\times 7^{2}$, 64 stride 1, $2^{2}$ & maxpool $1\times 3^{2}$ stride 1, $2^{2}$	conv $1\times 7^{2}$, 32 stride 1, $1^{1}$	RGB: $8\times 56^{2}$ Pose: $32\times 56^{2}$
res1	$\left[\begin{array}{c} 1\times 1^{2},64\\ 1\times 3^{2},64\\ 1\times 1^{2},256 \end{array}\right]\times 3$	N.A.	RGB: $8\times 56^{2}$ Pose: $32\times 56^{2}$
res2	$\left[\begin{array}{c} 1\times 1^{2},128\\ 1\times 3^{2},128\\ 1\times 1^{2},512 \end{array}\right]\times 4$	$\left[\begin{array}{c} 1\times 1^{2},32\\ 1\times 3^{2},32\\ 1\times 1^{2},128 \end{array}\right]\times 4$	RGB: $8\times 28^{2}$ Pose: $32\times 28^{2}$
res3	$\left[\begin{array}{c} \underline{3\times 1^{2}},256\\ 1\times 3^{2},256\\ 1\times 1^{2},1024 \end{array}\right]\times 6$	$\left[\begin{array}{c} \underline{3\times 1^{2}},64\\ 1\times 3^{2},64\\ 1\times 1^{2},256 \end{array}\right]\times 6$	RGB: $8\times 14^{2}$ Pose: $32\times 14^{2}$
res4	$\left[\begin{array}{c} \underline{3\times 1^{2}},512\\ 1\times 3^{2},512\\ 1\times 1^{2},2048 \end{array}\right]\times 3$	$\left[\begin{array}{c} \underline{3\times 1^{2}},128\\ 1\times 3^{2},128\\ 1\times 1^{2},512 \end{array}\right]\times 3$	RGB: $8\times 7^{2}$ Pose: $32\times 7^{2}$

**Table 3 sensors-24-07682-t003:** The evaluation of PoseTransformer3D. The evaluation involves comparing the parameters (Parms) and floating-point operations (FLOPs) of MS-G3D, Pose-SlowOnly, and PoseTransformer3D.

Dataset	MS-G3D			Pose-SlowOnly			PoseTransformer3D		
	Acc (%)	Params	FLOPs	Acc (%)	Params	FLOPs	Acc (%)	Params	FLOPs
FineGYM	92.0	2.8 M	24.7 G	93.2			93.8		
NTU RGB+D 60-X-Sub	91.9	2.8 M	16.7 G	93.7	2.0 M	15.9 G	94.1	4.0 M	17.7 G
NTU RGB+D 120-X-Sub	84.8	2.8 M	16.7 G	86.0			86.7		

**Table 4 sensors-24-07682-t004:** Comparison of the proposed PoseTransformer3D with PoseConv3D. The inputs are joint/limb-based 3D heatmap volumes and the results obtained from 1-/10-clip testing and 3D heatmap volumes with dimensions of 48×56×56.

Dataset	Joint-Based (%)			Limb-Based (%)		
	PoseConv3D	PoseTransformer3D		PoseConv3D	PoseTransformer3D	
	10-Clip	1-Clip	10-Clip	10-Clip	1-Clip	10-Clip
FineGYM	93.5	93.5	93.9	93.6	93.3	94.0
HMDB51	69.4	69.8	70.5	-	-	-
NTU RGB+D 60-C-Sub	93.7	93.8	94.1	93.4	93.5	93.8
NTU RGB+D 60-C-View	96.6	96.5	96.7	96.0	95.9	96.3
NTU RGB+D 120-X-Sub	86.0	86.6	86.7	85.9	86.2	86.8
NTU RGB+D 120-X-Set	89.6	90.0	90.1	89.7	90.0	90.1

**Table 5 sensors-24-07682-t005:** The results of PoseTransformer3D in late fusion. The outcomes obtained in 1-/10-clip testing using RGB volumes with dimensions of 8×224×224 and 3D heatmap volumes with dimensions of 32×56×56.

Dataset	RGB (%)	Pose (%)	Late Fusion (%)
NTU RGB+D 60-C-Sub	95.2/95.7	93.4/93.7	96.3/96.5
NTU RGB+D 60-C-View	98.6/99.0	96.5/96.7	99.1/99.2
NTU RGB+D 120-X-Sub	91.0/92.1	85.6/86.1	92.5/93.1
NTU RGB+D 120-X-Set	91.7/92.3	89.4/89.8	94.2/94.6

**Table 6 sensors-24-07682-t006:** The results of RGB-PoseTransformer3D in early + late fusion. We utilize temporal lengths of 8 for the RGB pathway and 32 for the pose pathway.

Dataset	RGB (%)	Pose (%)	Early + Late Fusion (%)
NTU RGB+D 60-C-Sub	96.8/97.0	96.8/96.9	97.0/97.1
NTU RGB+D 60-C-View	99.4/99.5	99.3/99.3	99.6/99.6
NTU RGB+D 120-X-Sub	94.2/94.5	94.0/94.5	94.2/94.7
NTU RGB+D 120-X-Set	96.1/96.3	96.1/96.2	96.1/96.3

**Table 7 sensors-24-07682-t007:** Comparison of the proposed PoseTransformer3D to the state-of-the-arts. The first section shows are utilized: “joint stream” and “bone stream” based methods. The second section includes joint (J) based heatmaps which denote using the same human skeletons as ours. The third section refers to the use of joints (J) and limbs (L) to adopt the method of late-stage fusion. Numbers with * are reported by [15].

Method	NTU RGB+D 60	NTU RGB+D 120	FineGYM (%)
	C-Sub/C-View (%)	X-Sub/X-Set (%)	
ST-GCN [33]	81.5/88.3	70.7/73.2	25.2 *
AS-GCN [41]	86.8/94.2	78.3/79.8	-
EfficientGCN-B0 [42]	89.9/94.7	85.9/84.3	-
MS-G3D [40]	91.5/96.2	86.9/88.4	-
TD-GCD [43]	92.8/96.8	-	-
PG-GCN [44] (*J*)	88.1/90.9	82.7/84.2	-
MS-G3D [40] (*J*)	92.2/96.6	87.2/89.0	-
Pose-C3D [6] (*J*)	93.1/97.0	86.0/89.6	93.2
Ours (*J*)	94.1/96.7	86.7/90.1	93.9
PG-GCN [44] (J+L)	91.8/95.8	88.4/88.8	-
Pose-C3D [6] (J+L)	94.1/97.1	86.9/90.3	94.3
Ours (J+L)	94.3/97.1	87.4/90.8	94.3

**Table 8 sensors-24-07682-t008:** Comparison of the proposed RGB-PoseTransformer3D to the state-of-the-arts. Notation of the header: RGB: R, depth: D, Skeleton: S, pose: P, and pose indicate joint-based heatmaps.

Method	Modality	NTU RGB+D 60	NTU RGB+D 120
		C-Sub/C-View (%)	X-Sub/X-Set (%)
SC-ConvNets [45]	R + D	89.4/91.2	86.9/87.7
SFMCB [46]	R + D	91.3/92.8	89.3/90.3
VPN [47]	R + S	95.5/98.0	86.3/87.8
3DA [48]	R + S	94.3/97.9	90.5/91.4
STAR-Transformer [49]	R + S	92.0/96.5	90.3/92.7
HAC [50]	R + S	95.7/98.8	93.7/94.5
MMNet [51]	R + S	96.0/98.8	92.9/94.4
PoseConv3D [6]	R + P	97.0/99.6	95.3/96.4
Ours	R + P	97.2/99.6	95.4/96.6
Ours+PoseConv3D	R + P	97.3/99.6	95.5/96.6

**Table 9 sensors-24-07682-t009:** Comparison of pre-training with no pretraining in 1-clip testing.

Dataset	N Pre-Training (%)	Pr-Training (%)
FineGYM	91.5	93.5
HMDB51	62.0	69.8
NTU RGB+D 60-C-Sub	93.1	93.8
NTU RGB+D 60-C-View	95.4	96.5
NTU RGB+D 120-X-Sub	85.2	86.6

**Table 10 sensors-24-07682-t010:** Ablation experiment on the number of global transformer decoder layers in PoseTransformer3D on the NTU RGB+D 60 dataset.

♯ of Decoders	C-Sub (%)	C-View (%)
No self-attention	94.9/95.4	97.5/98.1
1	95.2/95.7	98.6/99.0
2	95.3/95.8	98.6/99.0
3	95.4/95.8	98.7/99.1

**Table 11 sensors-24-07682-t011:** Ablation experiment on the number of queries in PoseTransformer3D on the FineGYM dataset.

♯ of Queries	FineGYM (%)
No self-attention	93.5
1	93.9
20	93.9
50	93.9
100	94.0

**Table 12 sensors-24-07682-t012:** Ablation study on the structure of global cross blocks using the FineGYM dataset.

Design	Joint (%)	Limb (%)
Convolution	93.5	93.6
Attention	93.7	93.7
Global MHRA	93.7	93.8
Global MHRA + DPE	93.9	94.0

**Table 13 sensors-24-07682-t013:** Ablation study conducted to analyze the structure of fusion for pose and RGB features using the NTU RGB+D 60 dataset.

Design	C-Sub (%)	C-View (%)
Late fusion	96.3/96.5	99.1/99.2
Late fusion + GCCB	97.0/97.1	99.6/99.6

## Data Availability

The datasets analyzed during the current study are available at https://rose1.ntu.edu.sg/dataset/actionRecognition/ (accessed on 1 October 2024). The datasets used in this study are all public datasets, and this manuscript does not contain any information about the study participants.

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
