# Peer review of "Empowering Efficient Spatio-Temporal Learning with a 3D CNN for Pose-Based Action Recognition"

_sensors, 2024, doi:10.3390/s24237682_

Round 1
Reviewer 1 Report
Comments and Suggestions for Authors
This letter includes my comments on the article " Empowering efficient spatio-temporal learning to 3D CNN for pose-based action recognition". In this paper, The authors propose a new attitude-based action recognition method, which uses 3D heatmap volumes and Global Cross Blocks (GCB) to extract spatiotemporal features, and performs multimodal feature learning through the RGB-Pose Transformer 3D framework. This new method can improve the efficiency and accuracy of gesture-based action recognition, and its effectiveness is verified by experiments. This study may provide guidance and recommendations for future research.
However, the paper needs a thorough review and significant changes in order to be accepted. Here are some of the reasons:
1. The authors propose a new model for behavior recognition, but there is a lack of a large number of practical test data to verify the feasibility of the model.
2. The caption text in Figure 1 of the main text is too small to be read clearly.
3. Although there are comparisons with existing methods, there is a lack of direct comparisons with current state-of-the-art models.
4. Any model has its limitations, and the article can discuss the limitations of the model and possible directions for improvement.
5. In the Introduction section, some closely related literature should be included in the revised version, such as the following: Spectrochimica Acta Part A: Molecular and Biomolecular Spectroscopy,2024, 311, 124038, Anal. Chem., 2022, 94(29), 10462-10469.
Author Response
1. The authors propose a new model for behavior recognition, but there is a lack of a large number of practical test data to verify the feasibility of the model.
Response 1:
Thank you very much for your review comments. We strongly agree with your point that having a large number of practical test data is very important for action recognition models. To address this challenge, we pay special attention to the selection of datasets during our research. Specifically, we use four widely recognized standard video datasets as the basis for our experiments. And, at the end of this paper, we added the next step of research, which is to apply it to practical situations.
In the revised manuscript:
(Lines 433-435)
“…However, in this paper, we only conducted testing on public datasets. In the future, lightweight design of models and testing in real-world scenarios can help achieve more efficient and practical value.”
2. The caption text in Figure 1 of the main text is too small to be read clearly.
Response 2:
Thank you very much for your review comments. We have carefully considered your suggestions and made every effort to improve the title text of Figure 1 in the main text.
3. Although there are comparisons with existing methods, there is a lack of direct comparisons with current state-of-the-art models.
Response 3:
Thank you very much for your review comments. We strongly agree with your point that direct comparison with current state-of-the-art models is crucial to evaluate the performance of our proposed method. In our study, we have conducted a detailed comparison with the current state-of-the-art models on the NTU RGB+D 60 and NTU RGB+D 120 dataset. And, the comparison results are shown in Tables 7 and 8.
4. Any model has its limitations, and the article can discuss the limitations of the model and possible directions for improvement.
Response 4:
Based on your suggestions, we have added the related information in the revised manuscript. The concrete revisions are as follows:
In the revised manuscript:
(Lines 332-339)
“…Despite PoseTransformer3D strengths, the computational complexity of the model is significant, with increased parameters and FLOPs that may limit real-world deployment, particularly in resource-constrained environments. To effectively address this challenge, we can adopt a model pruning strategy to reduce the number of parameters and computation. Specifically, we optimized the number of convolutional network layers from 4 to 3. At the same time, the GCB network was also adjusted to adopt a single-layer self-attention mechanism. These improvements significantly reduced the complexity of the model and improved its feasibility in practical applications. …”
5. In the Introduction section, some closely related literature should be included in the revised version, such as the following: Spectrochimica Acta Part A: Molecular and Biomolecular Spectroscopy,2024, 311, 124038, Anal. Chem., 2022, 94(29), 10462-10469.
Response 5:
Thank you very much for your review comments. We have added closely related literature to the paper.
In the revised manuscript:
(Lines 16-20)
“Action recognition plays a crucial role in video understanding, and previous research has investigated multiple modalities for feature representation, including RGB, skeleton, depth, and Point Cloud [1-4]. Specially, the skeleton-based methods are concise and effective for action recognition due to their action-focusing nature and compactness….”
Reviewer 2 Report
Comments and Suggestions for Authors
This article it's a great example of a well written and scientifically sound project which I encourage to be published on its present form
Author Response
This article it's a great example of a well written and scientifically sound project which I encourage to be published on its present form
Response: Thank you very much for the recognition of our research results by the reviewers. Your encouragement is our greatest motivation.
Reviewer 3 Report
Comments and Suggestions for Authors
The article presents a PoseTransformer3D framework that combines 3D CNNs with Global Cross Blocks (GCB) to capture spatio-temporal dependencies. In addition, the RGB PoseTransformer3D enhances multimodal feature fusion and achieves state-of-the-art results on benchmarks such as NTU RGB+D and FineGYM.
Strength:
The proposed method addresses limitations of existing pose-based action recognition frameworks by integrating self-attention mechanisms with 3D CNNs. The use of GCB and GCCB enables effective learning of both local and global dependencies, while the multimodal RGB-PoseTransformer3D framework demonstrates performance in fusing RGB and pose data. The experimental setup is robust, with extensive evaluations, ablation studies and comparisons validating the method. The article also includes clear visualizations and detailed explanations of the proposed architecture.
Shortcomings:
Despite its strengths, the computational complexity of the model is significant, with increased parameters and FLOPs that may limit real-world deployment, particularly in resource-constrained environments. The reliance on pre-training raises concerns about applicability to resource-constrained scenarios. The evaluation of the paper focuses primarily on accuracy, overlooking important practical metrics such as inference speed and robustness to noisy inputs. Minor redundancies in the text and occasional grammatical problems detract slightly from clarity. In addition, qualitative analyses, such as visualizations of error cases or model attention mechanisms, are limited.
Conclusion:
The article makes a contribution to the field of action recognition and meets the standards for Sensors journal (Q1). However, addressing the minor shortcomings outlined would increase its impact and applicability. I recommend minor revisions.
Comments on the Quality of English LanguageThe English could be improved to more clearly express the research.
Author Response
1. Despite its strengths, the computational complexity of the model is significant, with increased parameters and FLOPs that may limit real-world deployment, particularly in resource-constrained environments.
Response 1:
Based on your suggestions, we have added the related information in the revised manuscript. The concrete revisions are as follows:
In the revised manuscript:
(Lines 332-339)
“…Despite PoseTransformer3D strengths, the computational complexity of the model is significant, with increased parameters and FLOPs that may limit real-world deployment, particularly in resource-constrained environments. To effectively address this challenge, we can adopt a model pruning strategy to reduce the number of parameters and computation. Specifically, we optimized the number of convolutional network layers from 4 to 3. At the same time, the GCB network was also adjusted to adopt a single-layer self-attention mechanism. These improvements significantly reduced the complexity of the model and improved its feasibility in practical applications. …”
2. The reliance on pre-training raises concerns about applicability to resource-constrained scenarios. The evaluation of the paper focuses primarily on accuracy, overlooking important practical metrics such as inference speed and robustness to noisy inputs.
Response 2:
Based on your suggestions regarding pre-training, we have added the related information in the revised manuscript. The concrete revisions are as follows:
In the revised manuscript:
(Lines 382-387)
“…Pose-Transformer has to require tedious and complex image pre-training phase before it can be fine-tuned to adapt to video processing tasks. This blocks its wide usage in practice. In contrast, open-sourced CNNs are readily available and well pretrained with rich image supervision. Based on these observations, we propose a generic paradigm to build our networks, by combining a pre-trained CNN with an efficient GCB. Pose-Transformer can achieve faster convergence speed and lower computing resource requirements. …”
To address the robustness issue, we adopt a dual-stream network architecture. RGB flow can capture rich details and texture information, while Pose flow effectively removes noise interference, thereby significantly improving the robustness and stability of the model in complex environments.
3. Minor redundancies in the text and occasional grammatical problems detract slightly from clarity.
Response 3:
Thank you very much for your review comments. We have carefully considered the suggestions you have made and made every effort to improve the language quality of the paper. We have carefully checked the grammar and style in the paper to ensure accuracy and compliance with academic writing standards.
4. In addition, qualitative analyses, such as visualizations of error cases or model attention mechanisms, are limited.
Response 4:
We conducted a detailed qualitative analysis of the self-attention mechanism of the model. In Table 11, we listed the qualitative analysis results of the number of self-attention mechanism layers in detail. In Table 12, we conducted an in-depth analysis of each structure of the GCB network to ensure a comprehensive understanding of the role. Based on your suggestions, we have added the related information in the revised manuscript. The concrete revisions are as follows:
In the revised manuscript:
(Lines 411-414)
“…In Figure 7, we can find a visualization of the prediction accuracy for each category. We qualitatively analyze the error categories and find is limited in its ability to distinguish between highly similar actions. …”